# Adaptive Method to Reduce Thermal Deformation of Ball Screws Based on Carbon Fiber Reinforced Plastics

**DOI:** 10.3390/ma12193113

**Published:** 2019-09-24

**Authors:** Xiangsheng Gao, Zeyun Qin, Yueyang Guo, Min Wang, Tao Zan

**Affiliations:** Beijing Key Laboratory of Advanced Manufacturing Technology, College of Mechanical Engineering and Applied Electronics Technology, Beijing University of Technology, Beijing 100124, China

**Keywords:** ball screw, thermal deformation, carbon fiber reinforced plastics, finite element analysis

## Abstract

In high-speed precision machining, thermal deformation caused by temperature rise affects the accuracy stability of the machine tool to a significant extent. In order to reduce the thermal deformation of ball screws and improve the accuracy, a new adaptive method based on carbon fiber reinforced plastics (CFRP) was proposed in this study and the thermal deformation of ball screws was determined. By using the sequential coupling method, the thermal–structural coupling analysis of a ball screw was conducted based on the finite element method (FEM). The analysis results were verified through a comparison with the experimental results. Based on the verification, an FE model of the improved ball screw was established to study its thermal characteristics. The key design parameters of the improved ball screw were optimized based on the Kriging model and genetic algorithm (GA). The thermal reduction effect of the improved ball screw was validated through the experimental results. The results indicate that the adaptive method proposed in this research is effective in reducing the thermal deformation of ball screws.

## 1. Introduction

Ball screws with high efficiency, precision, and stiffness have become important transmission mechanisms of the servo feed system used for transforming rotational motion into linear motion [1]. With the increasing popularity of high-speed machining technology, ball screw speeds are also increasing, and can reach 90~120 m/min [2,3]. The application of such high-speed ball screws has greatly improved the machining efficiency. However, they also face several issues such as the occurrence of thermal deformation, which can severely affect the precision of the ball screw [4]. In the feed system of machine tools, the heat induced by friction from the screw and nut, supporting bearing, motor, and other components may cause thermal deformation, particularly in high-speed precision machine tools. Thermal deformation of the ball screw will lead to a decrease in precision stability and significantly influence the machining precision. It has been reported that the thermal error of machine tool accounts for about 30%~50% of the total error, and this proportion can reach up to 70% in precision machining [5]. Therefore, better thermal performance of the ball screw is required. 

Determination of the thermal deformation of ball screws is an important prerequisite for reducing such unfavorable effects. Shi and Mei et al. established a thermal deformation model of the ball screw according to experimental data by considering the time and nut position based on the multiple regression, fuzzy serial, and linear regression methods [6,7]. Yang and Yin et al. [8] studied the thermal and dynamic characteristics of a high-speed spindle by the finite element (FE) method. The softening of support stiffness caused by the bearing centrifugal force and stiffness hardening caused by the thermal pre-tightening were considered in the analysis. Chen and Liu [9] established a thermal-structural model of angular contact ball bearings considering the effect of thermal response and pre-tightening mode of the system, which could effectively predict and control the thermal performance of the spindle during runtime and their influence on the dynamic characteristics of the spindle.

Thermal deformation compensation of the ball screw is an effective approach to reduce the error in machine tools. Some scholars established the temperature field and thermal deformation model of machine tools based on thermal modal analysis [10] or temperature collection [11], and the robustness of the model was verified by experiments. Based on the results obtained thereby, the thermal deformation was compensated for in real time [12,13]. Creighton et al. [14] developed an exponential growth model that could provide thermal deformation compensation for spindles in machine tools. Gómez-Acedo et al. [15] proposed a methodology for a thermal deformation compensation system, which could achieve higher precision levels in common shop floor environments. Xu and Liu [16] analyzed the thermal characteristics of a hollow ball screw by the FE method and the improved lumped heat capacity method. It was shown that the hollow design can reduce the temperature rise and obtain thermal balance quickly. In order to reduce the thermal deformation and improve the accuracy of machine tools, some components of the spindle are fabricated using advanced materials such as carbon fiber reinforced plastics (CFRP), instead of traditional steel. Such materials are not temperature-sensitive and can reduce the thermal deformation [17]. The carbon fiber in CFRP also has the advantage of high elastic modulus and strength for the application of thermal deformation reduction. Hamed Khoshnevis et al. [18] studied the mechanical behavior of carbon nanotube fibers spun from the floating catalyst method [19]. Effects of the alignment and compaction of carbon nanotube bundles on the time-dependent behaviors and relaxation process were investigated. Uhlmann and Marcus [20] proposed an active thermal deformation compensation method for a spindle based on CFRP, where carbon fibers were mounted in the circumferential direction along the outside of the spindle. As carbon fiber has a negative thermal expansion coefficient, carbon fibers reduce in length when they are heated using a thermal element, thus contracting the spindle and reducing its thermal deformation [21]. Ge and Ding [22] designed an improved spindle based on the negative thermal expansion coefficient of CFRP. The results of the simulation and experiments showed that the thermal deformation of the improved motorized spindle decreased by 97% compared with that before the improvement. It shows that the CFRP has good prospects in the field of thermal deformation reduction. However, the design used in the spindle system cannot be directly adapted to ball screws, because the position error of the nut in ball screws is always concentrated in the engineering practice. The thermal deformation at the nut varies along with the position of the nut during the actual operation. The thermal error compensation method in previous studies [21,22] is not suitable for ball screws, and an adaptive method for reducing the thermal deformation of ball screws is required.

In summary, there are two major strategies for the compensation of thermal deformation in a ball screw, namely, an active control system and a hollow (liquid or air cooled) ball screw. An active control system requires a complex system and the speed of the nut is much greater than the heat transfer speed, which can lead to failure of the compensation system. Hollow and liquid-cooled ball screws have a good compensation effect [23,24], but the hollow design reduces the stiffness of ball screws. Therefore, an adaptive thermal deformation reduction method of ball screws based on CFRP is proposed in this research. 

The contents of this paper are arranged as follows. In Section 2, an adaptive method to reduce the thermal deformation of the ball screw based on CFRP is proposed. In Section 3, FE simulation on ball screws is conducted and validated. On that basis, the key design parameters of improved ball screws are optimized. In Section 4, experimental studies are conducted on the improved ball screws to verify the effectiveness of this method. In Section 5, the conclusions of this research are presented.

## 2. Adaptive Method to Reduce Thermal Deformation of Ball Screw

In order to reduce the thermal deformation of ball screws, an improved ball screw was designed by utilizing the negative thermal expansion coefficient of CFRP in the axial direction, as illustrated in Figure 1. It should be noted that the direction of the fiber is parallel to the axial direction of the ball screw. When the temperature rises, the screw will expand in the axial direction. However, CFRP will contract the screw and reduce the thermal deformation. 

The design at the end is to ensure that the CFRP is in a tension state by adjusting the nut for pretension when it is in operation. The end caps are tightly connected by a series of bolts. When the ball screw feeding system is assembled, nuts and bearings in the ball screws are assembled first, then the CFRP is mounted, and finally the end cap is installed, as shown in Figure 2.

## 3. FE Simulation and Optimization

In order to check the effect and obtain better performance, thermal deformation of the ordinary ball screw and improved ball screw were simulated and compared, and the FE modeling was validated by experimental results. On that basis, the key design parameters were optimized.

### 3.1. Determination of Thermal Deformation of Ball Screw

A ball screw can be approximated as a typical thermodynamic system, and the relevant thermodynamic parameters such as heat generation and heat transfer coefficient of the screw surface can be obtained from the literature [25,26,27,28,29,30]. There are two main heat sources in a ball screw drive system: the nut-screw interface and support bearings. A numerical simulation was conducted when the rotational speed was 1000 r/min. The comparison between the model with heat generation from the support bearings and that without heat generation from the support bearings revealed that the maximum proportion of supporting bearings effect was 14.6%. Therefore, the heat generation from support bearings was ignored in this research. 

There are two typical methods for evaluating the thermal deformation of ball screws based on the FE method. The first is the sequential coupling method where the interaction between the temperature field and the structural field of ball screw is neglected. The results of the thermal analysis are directly transferred into the structural field as a temperature load, and the thermal deformation is obtained using the structural analysis. In the second method, the interaction between the two physical fields is considered. According to previous research [31], there is a little difference in the results between the two methods, but the second method is more computationally expensive and requires more time for the solution. Therefore, a sequential coupling method was adopted to obtain the thermal deformation. 

In order to illustrate the FE modeling of ball screws, a certain type of ball screw was investigated. Parameters of the ball screw are given in Table 1. The ball screw was mounted on the test table, as shown in Figure 3, and no axial load was applied on the worktable. The slave nut was designed to measure the friction torque of the ball screws. In this study, the rotational speed of the ball screw was 1000 r/min.

During geometric modeling, complex features of the ball screw such as the raceway were ignored, and the screw was simplified to a shaft, thereby reducing the demand for computing resources and time [6]. The FE model was established using solid elements in ANSYS software as shown in Figure 4.

Thermal parameters of the ball screw are determined as shown in Table 2. The forced convection occurs on the outer surface of the ball screw. The APDL (ANSYS Parametric Design Language) command is programmed to simulate the reciprocating cycle of heat flow according to the movement of the nuts. As shown in Figure 5, at the moment *t*, the heat flow is applied in the blue region, while at *t* + Δ*t*, the applied region moves a distance toward the nut moving direction. The distance can be evaluated by *v*
× Δ*t*, where *v* is the velocity of the nut movement. When the region moves to the end, it will move back until the thermal balance is achieved. The trace of nut motion is shown in Figure 6. It can be seen that the heat flow is not applied on all nodes in the model, and it is never applied on some nodes in the middle and at the end of the ball screw.

The steady-state temperature field, shown in Figure 7a, and the temperature rise of typical node of the ball screw, namely Node A shown in Figure 7b, were obtained from the results of the thermal analysis. It can be seen that the temperature of Node A rose rapidly at the beginning, slowed down with time, and finally became stable. Node A had the maximum temperature in the system.

During the structural solution, the bearing close to the motor in the feeding system restrains the axial deformation of the ball screw, while the bearing in the opposite direction provides the freedom for the axial deformation. Therefore, based on its working conditions, the ball screw is considered to be a simply supported beam. Based on the sequential coupling method, the displacement field of the ball screw was obtained as shown in Figure 7c. As seen from the figure, the displacement of the ball screw increased uniformly along the ball screw. 

### 3.2. Experimental Verification

In order to verify the FE model of the ball screw, the experiment was carried out on the ball screw test table, as shown in Figure 8a. A thermal imager (Ti55FT, FLUKE, Everett, USA) was employed to measure the temperature field of the ball screw and the thermal deformation of the ball screw was measured by a dial gauge, as shown in Figure 8b.

The experiment was carried out without applying any axial load and only the preload was considered. The rotational speed was 1000 r/min. The thermal imager measured the temperature of the ball screw once every 10 minutes. The experiment was conducted until the ball screw on the test table attained thermal balance. The steady-state field of the ball screw experiment is shown in Figure 9, and it can be observed that the obtained temperature distribution matched the result of the simulation. It can be concluded that the ball screw reached thermal balance in 70 minutes.

The comparison between the simulation results and experimental results is given in Table 3 and Figure 10. It can be observed that the maximum error of temperature was 17.3%, and the maximum error of thermal deformation was 18.6%. The FE model can be considered effective. However, the experimental results were higher than the FE results. The reason for the error was mainly due to the heat generated by the motor and other mechanical friction, which causes the value measured in the experiment to be slightly higher than that obtained in the simulation. 

### 3.3. FE Model of Improved Ball Screw

To study the reduction effect of the new design, FE analysis of the improved ball screw was performed. According to the method described in Section 3.1, the FE model of the improved ball screw was established using solid elements, as shown in Figure 11. In order to simulate the harmonization of deformation and temperature, the CFRP and screw shared the same node at the interface. It must be noted that there was only one nut in the model, which is different from that in Section 3.1 and Section 3.2. In this model, the following assumptions were made: hole diameter *d*_i_ = 25 mm, thickness of the layer *b* = 2.5 mm, and volume fraction of carbon fiber *c*_m_ = 65%. High modulus carbon fiber (M55J type) was used as the reinforcement of CFRP. The parameters of carbon fiber are shown in Table 4. The temperature and thermal deformation of the improved ball screw were obtained, as shown in Figure 12.

### 3.4. Key Parameters Optimization

The key design parameters of the improved ball screw are *d*_i_, *b,* and *c*_m_. The diameter of the hole affects the heat transfer coefficient; the thickness of CFRP layer affects the heat dissipation and contraction, while the volume fraction of the fiber affects the elastic modulus and thermal expansion coefficient of the composite.

A Kriging model was established to make the optimization more efficient. The orthogonal table *L*_16_ (4^3^) with three factors and four levels was employed to generate the samples, as shown in Table 5. According to the method for determining the thermal deformation of ball screws mentioned in Section 3.3, the maximum axial thermal deformation in each sample was determined, as shown in Table 6.

The results of the range analysis are shown in Table 7. It can be concluded that diameter *d*_i_ had the greatest effect on the thermal deformation of the improved ball screw. The effect of layer thickness on the thermal deformation of the improved ball screw was slightly greater than that of the fiber volume fraction. 

The Kriging model is established according to the samples [32], and the relationship between the maximum axial thermal deformation and key design parameters are given, which can be expressed as $u=\varphi (c_{m},di,b)$. Figure 13 shows the response surface for the thermal deformation of the improved ball screw based on the Kriging model. 

There are two important indicators to evaluate the Kriging model [33]. The average error represents the accuracy of the model in the feasible region, and the maximum error indicates the maximum extent to which the surrogate model deviates from the compared value. 

The average error is derived as $e_{a}=\frac{1}{n}\sum_{i=1}^{n}\left|\frac{y_{i}(x)-{\hat{y}}_{i}(x)}{y_{i}(x)}\right|$; the maximum error is $e_{m}=\max\left[\frac{y_{i}(x)-{\hat{y}}_{i}(x)}{y_{i}(x)}\right]$ (*i*=1,2, …*n*), where *y* and $\hat{y}$ are the FE value and predicted value of the samples, respectively. *n* is the number of samples.

To validate the Kriging model, eight samples were randomly selected and calculated by the FE method, as listed in Table 8.

The thermal deformation calculated from the Kriging model and FE method is shown in Figure 14. The average error was calculated as 6.26%, while the maximum error was calculated as 16.26%. The maximum error occurred at the 4^th^ verification sample. This is because the gradient at this sample (*c*_m_ = 55%, *d*_i_ = 16 mm, *b* = 3.5 mm) was much greater than that at other samples in the real response surface. The Kriging model was based on orthogonal designed samples, which resulted in the greater error at this point (*c*_m_ = 55%, *d*_i_ = 16 mm, *b* = 3.5 mm). Therefore, it can be concluded that the Kriging model developed in this study can be implemented to predict the thermal deformation of the improved ball screws instead of the FE method. 

The optimization of the design parameters for the improved ball screw was implemented based on genetic algorithm (GA) [34] with the thermal deformation being defined as the objective function.
(1)$$\varphi \left(c_{\mathrm{mo}},d_{\mathrm{io}},b_{o}\right)=\min\varphi \left(c_{m},d_{i},b\right)$$

The optimized results are shown in Table 9. The simulation results of the ordinary ball screw and optimized improved ball screw are shown in Figure 15. It can be observed that the axial thermal deformation of the optimized ball screw was 46.8 μm, which was 22.3% less than that of the ball screw before improvement. There was shear stress between ball screw and CFRP, as shown in Figure 16, indicating that the design and optimization were effective.

## 4. Experimental Study

In order to verify the effectiveness of the method, a fundamental experiment was carried out. A hollow shaft with CFRP pasted inside was used to simulate the improved ball screw. A simplified shaft with the same diameter and length was also manufactured to simulate the ordinary ball screw and compare the temperature and thermal deformation with the improved design. As it is difficult to measure and control the preload, it was difficult to determine the heat generation at the nut. In this research, the shaft was heated by heating sheets to simulate the heat generation when the nut moves, and the heat power was adjusted by the supply voltage. 

### 4.1. Experimental Procedure

As shown in Figure 17, high-temperature ceramic heating sheets (XH-RJ101012 type) (Xinghe, Suqian, China) were attached on the surface of the shaft. The heating sheet was powered by a DC stabilized power supply (HY1711-5S type) (Yaguang, Huaian, China). Three ultra-thin temperature sensors (KYW-TC type) (Kunlunyuanyang, Beijing, China) were also arranged on the surface of the shaft to measure the temperature. A temperature-circuit-detect instrument (Kunlunyuanyang, Beijing, China) was employed for temperature monitoring. A dial gauge (Haliang, Haerbin, China) was fixed at the end of the shaft to measure the thermal deformation. The experimental setup is depicted in Figure 18. The ordinary ball screw model and the improved one were tested under the same conditions and then compared. Only the ordinary ball screw model is shown in Figure 17, and only the test for the improved design is depicted in Figure 18. It must be noted that they also had the same arrangement for the temperature sensors, heating sheets and dial gauge, and the same experimental setup.

Two working conditions were considered in this research, namely, the supply voltage of 20 V and 30 V. The temperature and thermal deformation of the two specimens were tested and compared. When the temperature rise was less than 0.5 °C in 10 minutes, it was considered that the system had reached thermal balance. 

### 4.2. Experimental Results and Analysis

The temperature rise comparison of the specimens is shown in Figure 19. It can be seen that the temperature rise decreased by 35.3%, 20.0%, and 17.7% in the three locations when the supply voltage was 20 V, and by 51.1%, 9.6%, and 9.5% when the supply voltage was 30 V due to the improved design. It was also found that the improved design resulted in a rapid temperature balance, which demonstrates that this method can improve the thermal characteristics of the ball screws. 

Figure 20 shows the comparison of thermal deformation at different supply voltages. It can be observed that the thermal deformation decreased by 59.0% and 70.1% when the supply voltage was 20 V and 30 V, respectively. Thus, it can be concluded that this adaptive method can reduce the thermal deformation of ball screws. 

The improved design obviously reduced the temperature rise and thermal deformation according to the experimental results. CFRP contracted the screw in the axial direction with a rise in temperature due to the negative thermal expansion coefficient, which reduces the thermal deformation. 

## 5. Conclusions

A detailed design was proposed for reducing the thermal deformation of ball screws adaptively based on the negative thermal expansion coefficient of CFRP. The effectiveness of this proposed method was theoretically and experimentally validated. The following conclusions can be drawn from this study.

(1) A new adaptive method based on CFRP was proposed to reduce the thermal deformation in ball screws. It was demonstrated that the adaptive method is effective in reducing the temperature and thermal deformation of the ball screw.

(2) The temperature and thermal deformation of ordinary ball screw and improved design were solved by FEM. The optimization of the key parameters of the improved ball screw was also conducted. The effectiveness of the adaptive method was validated by theoretical analysis.

(3) Based on the experimental study, the adaptive method was verified. The thermal deformation was reduced by as much as 70.1%, which indicates that the adaptive method proposed in this research is effective in reducing the thermal deformation of ball screws.

A fundamental experiment on the ball screw model was conducted in this research. The proposed method will be validated under actual working conditions at the next stage of this research.

## Figures and Tables

**Figure 1 materials-12-03113-f001:**
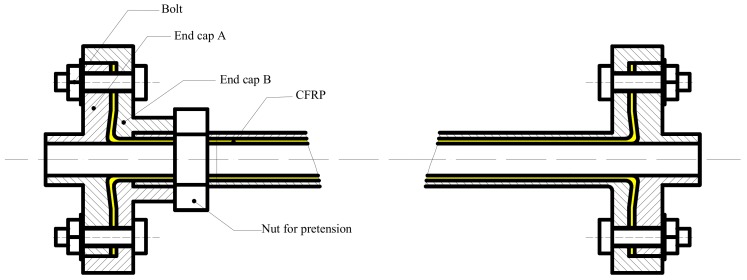
Schematic diagram of improved ball screw.

**Figure 2 materials-12-03113-f002:**
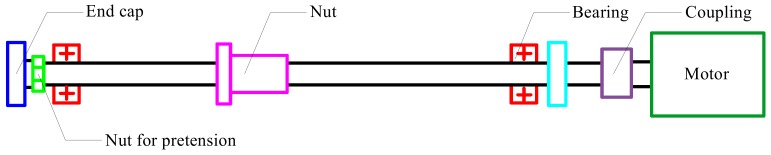
The improved ball screw assembly.

**Figure 3 materials-12-03113-f003:**
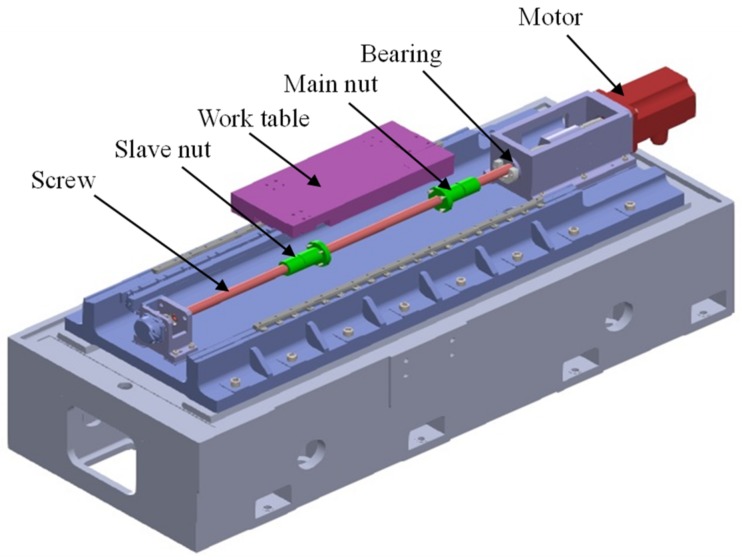
Test table for the ball screw.

**Figure 4 materials-12-03113-f004:**
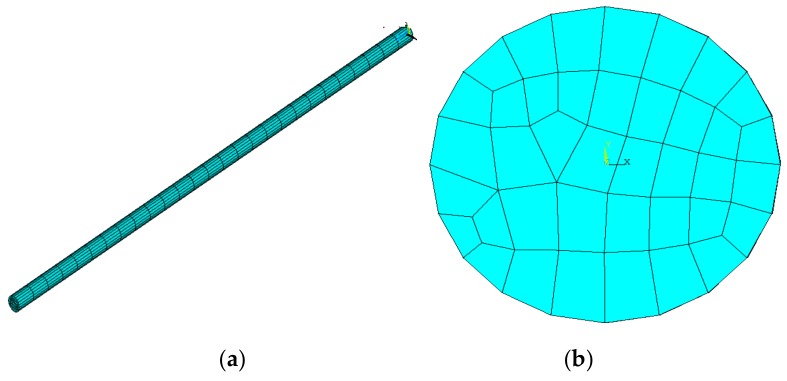
FE model (**a**) Isometric view; (**b**) Orthographic view.

**Figure 5 materials-12-03113-f005:**
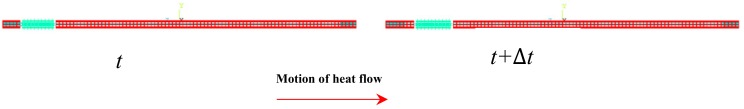
Heat flow motion in the FE method.

**Figure 6 materials-12-03113-f006:**
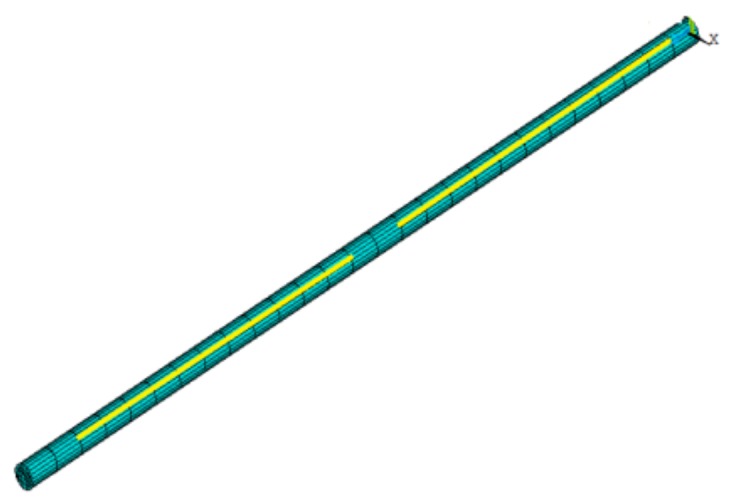
Trace of nut motion.

**Figure 7 materials-12-03113-f007:**
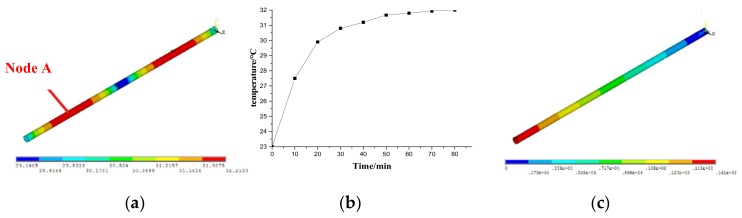
FE results of the ball screw. (**a**) Steady-state temperature field; (**b**) Temperature curve of Node A; (**c**) Displacement field.

**Figure 8 materials-12-03113-f008:**
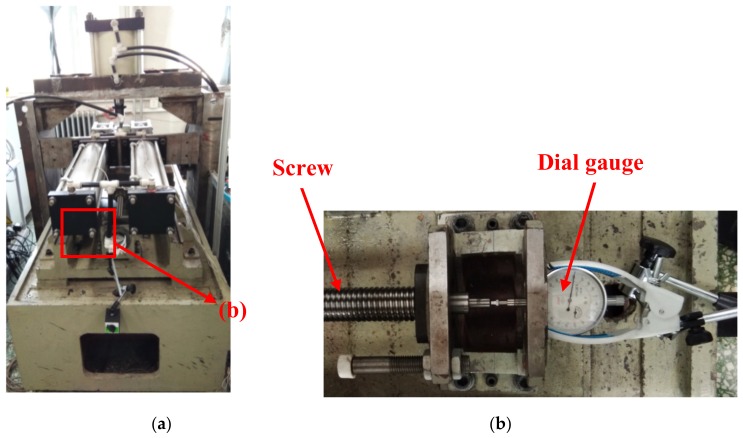
Experimental setup for thermal deformation measurement. (**a**) Whole view; (**b**) Detail view.

**Figure 9 materials-12-03113-f009:**
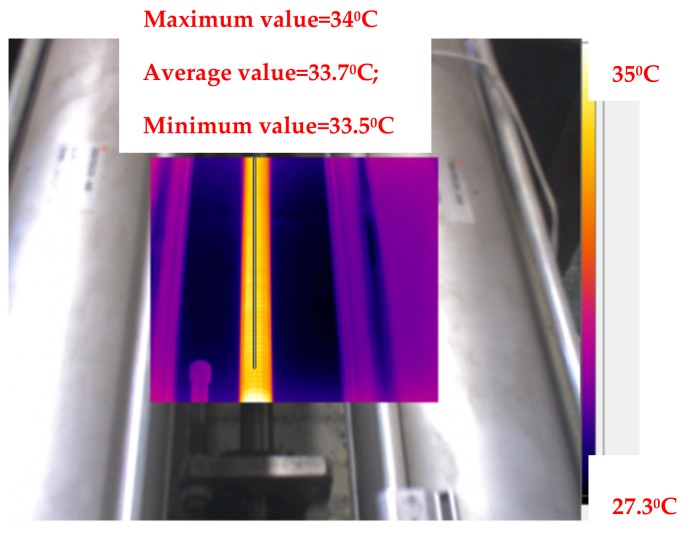
Experimental results: steady-state field of the ball screw.

**Figure 10 materials-12-03113-f010:**
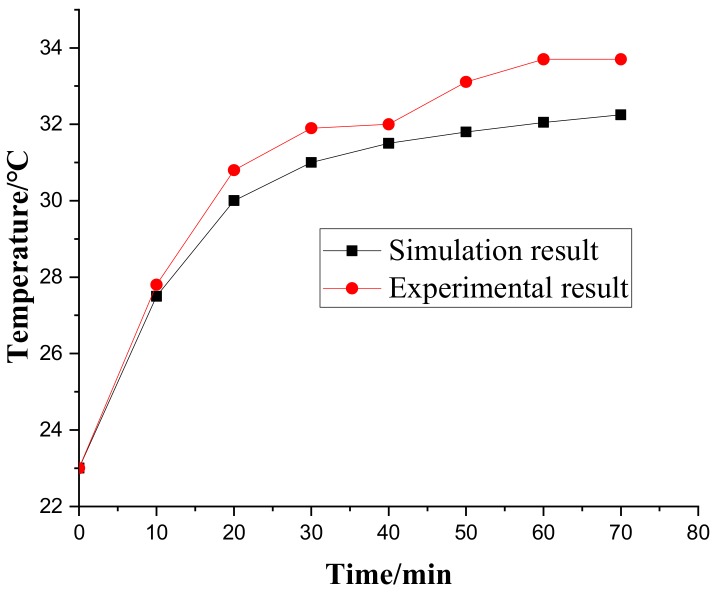
Temperature rise of Node A.

**Figure 11 materials-12-03113-f011:**
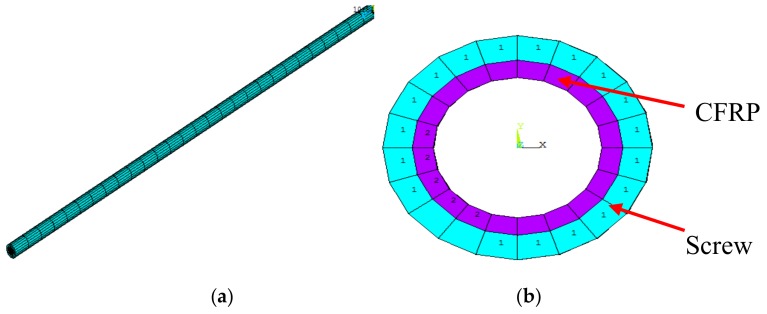
FE model (**a**) Isometric view; (**b**) Orthographic view.

**Figure 12 materials-12-03113-f012:**
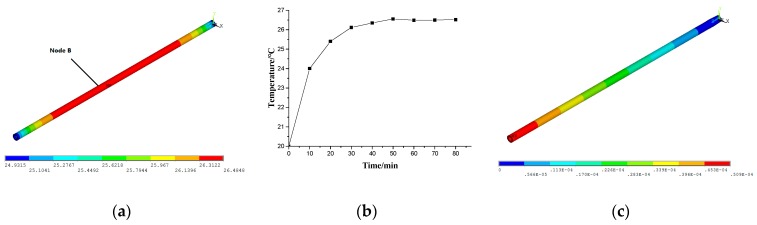
FE results of the improved ball screw. (**a**) Steady-state temperature field; (**b**) Temperature rise of Node B; (**c**) Displacement field.

**Figure 13 materials-12-03113-f013:**
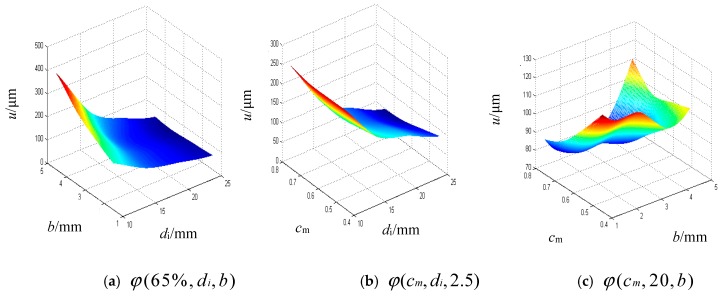
Response surfaces of the Kriging model.

**Figure 14 materials-12-03113-f014:**
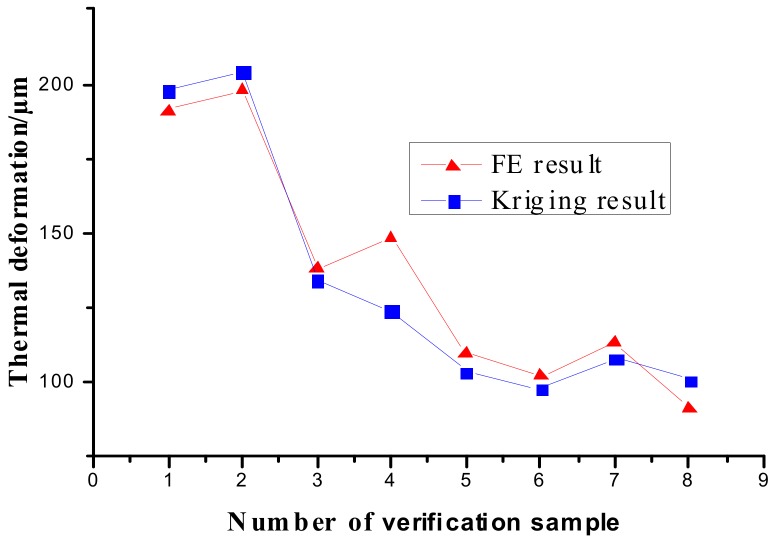
Comparison between the results obtained by different methods.

**Figure 15 materials-12-03113-f015:**
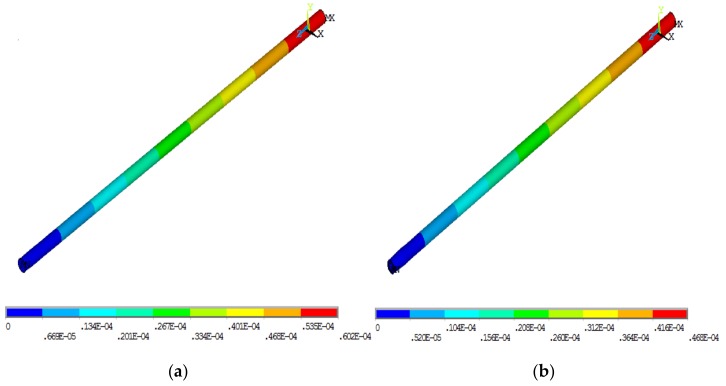
Simulation results. (**a**) Ordinary ball screw; (**b**) Optimized improved ball screw.

**Figure 16 materials-12-03113-f016:**
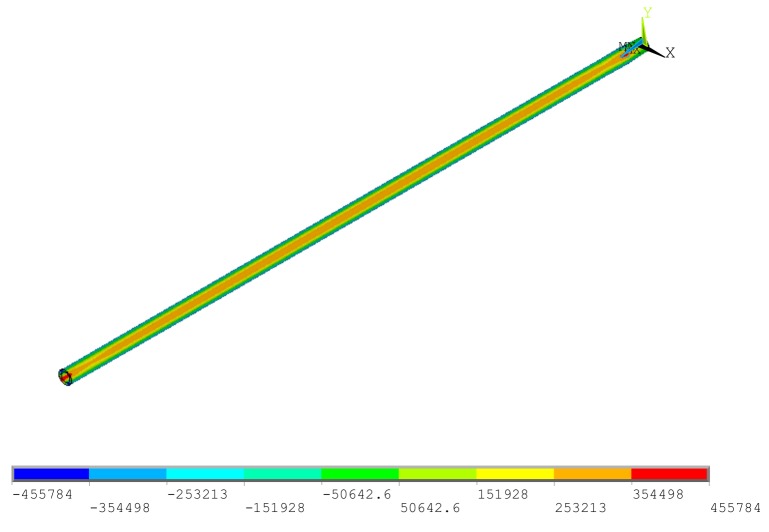
Shear stress distribution.

**Figure 17 materials-12-03113-f017:**
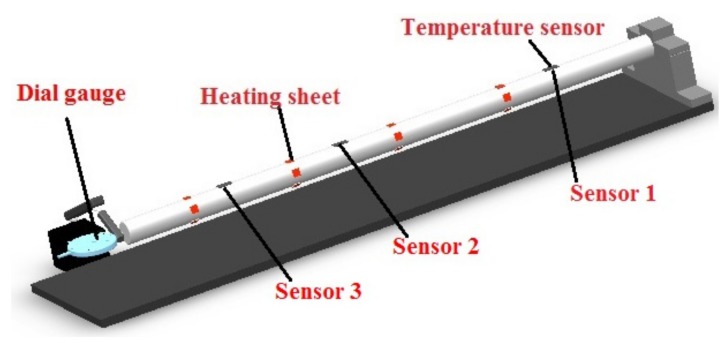
Location of the temperature sensors, heating sheets, and dial gauge.

**Figure 18 materials-12-03113-f018:**
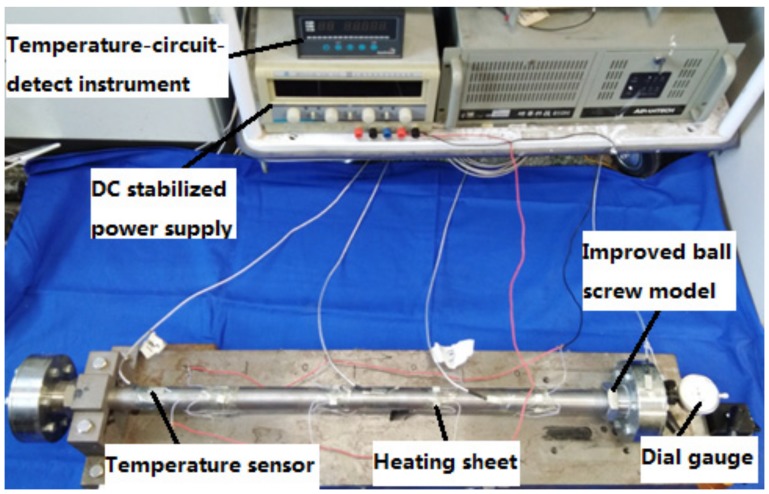
Experimental setup.

**Figure 19 materials-12-03113-f019:**
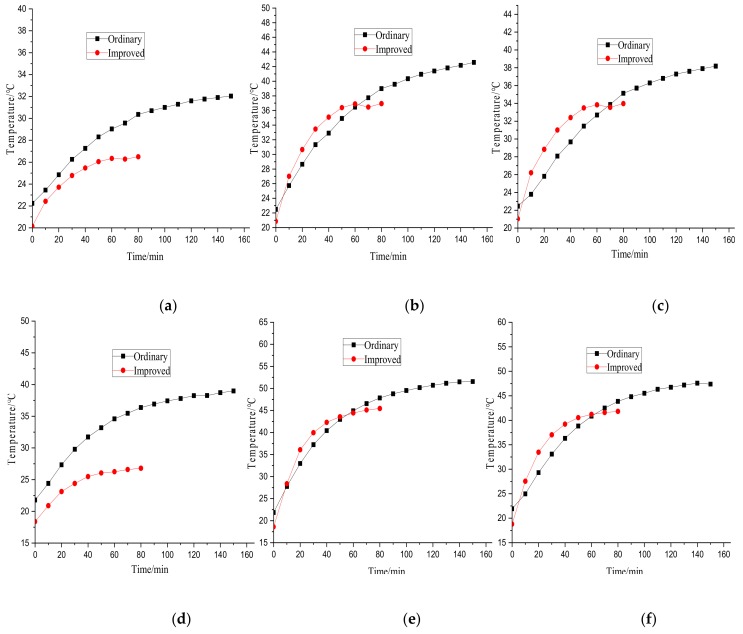
Temperature rise under different supply voltages. (**a**) Sensor 1, 20 V; (**b**) Sensor 2, 20 V; (**c**) Sensor 3, 20 V; (**d**) Sensor 1, 30 V; (**e**) Sensor 2, 30 V; (**f**) Sensor 3, 30 V.

**Figure 20 materials-12-03113-f020:**
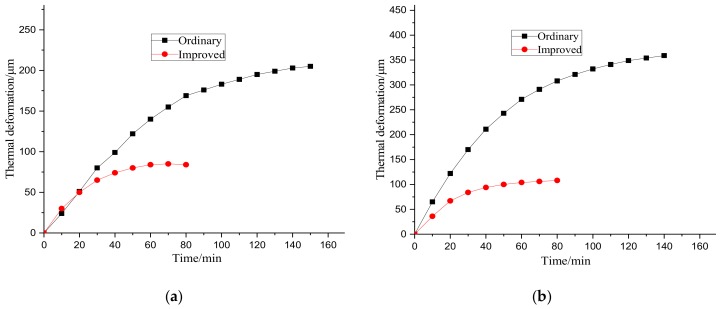
Deformation variation comparison. (**a**) 20 V; (**b**) 30 V.

**Table 1 materials-12-03113-t001:** Parameters of the ball screw.

Parameter	Value
Nominal diameter *d_o_*	32 mm
Ball diameter *D_b_*	5.953 mm
Lead *P_h_*	5 mm
Helix angle α	5.6833°
Rated dynamic load of ball screw *C*_a_	25628 N

**Table 2 materials-12-03113-t002:** Thermal parameters of ball screw.

Parameter	Value
Heat generation of nut *Q*	16.5 W
Heat transfer coefficient of outer surface *h*	38.4 W/(m^2^·K)
Thermal conductivity λ	40 W/(m·K)

**Table 3 materials-12-03113-t003:** Comparison of experimental and FE results.

	FE Results	Experimental Results	Error
Maximum temperature rise	9.2533 °C	11 °C	15.8%
Minimum temperature rise	7.524 °C	9.1 °C	17.3%
Thermal deformation	161 μm	198 μm	18.6%

**Table 4 materials-12-03113-t004:** Parameters of M55J.

Parameter	Value
Tensile modulus	540 GPa
Density	1.91 g/cm^3^
Thermal expansion coefficient	−1.1 × 10^−6^ m/K
Specific heat	711 J/(kg·K)
Thermal conductivity	155.72 W/m·K

**Table 5 materials-12-03113-t005:** Parameters of the improved ball screw.

*c* _m_	*d*_i_/mm	*b*/mm
45%	10	1.5
55%	15	2.5
65%	20	3.5
75%	25	4.5

**Table 6 materials-12-03113-t006:** Samples for the Kriging model.

No.	*c* _m_	*d*_i_/mm	*b*/mm	u/μm
1	45%	10	1.5	211
2	45%	15	2.5	150
3	45%	20	3.5	103
4	45%	25	4.5	60.5
5	55%	10	2.5	251
6	55%	15	1.5	132
7	55%	20	4.5	105
8	55%	25	3.5	53.5
9	65%	10	3.5	315
10	65%	15	4.5	189
11	65%	20	1.5	86.6
121	65%	25	2.5	50.9
13	75%	10	4.5	440
14	75%	15	3.5	154
15	75%	20	2.5	83.3
16	75%	25	1.5	51.9

**Table 7 materials-12-03113-t007:** Range analysis results.

	*c* _m_	*d* _i_	*b*
1st level	131.125	304.25	120.375
2nd level	135.375	156.25	133.8
3rd level	160.375	94.475	156.375
4th level	182.3	54.2	198.625
Range	51.175	250.05	78.25

**Table 8 materials-12-03113-t008:** Verification samples for the Kriging model.

*d*_i_/mm	*b*/mm	*c* _m_	u/μm
11	1.5	45%	191
12	2.5	55%	198
14	1.5	75%	138
16	3.5	55%	148
17	1.5	65%	110
18	2.5	75%	102
19	3.5	45%	113
20	2.5	55%	91.6

**Table 9 materials-12-03113-t009:** Optimized design parameters of the improved ball screw.

	*d*_io_/mm	*b*_o_/mm	*c* _mo_
Value	25	3.6	0.67

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
