# Peer review of "Adaptive Method to Reduce Thermal Deformation of Ball Screws Based on Carbon Fiber Reinforced Plastics"

_materials, 2019, doi:10.3390/ma12193113_

Round 1

Reviewer 1 Report

The authors report an adaptive method based on carbon fiber reinforced plastics (CFRP) to reduce the thermal deformation of ball screws and improve its accuracy. To study the thermal properties, experimental data regarding FEM of the improved ball screw has been represented appropriately. However, there are a number of typos and grammar errors in the submitted manuscript. For this reason, I recommend this manuscript to be published in Materials journal after an appropriate editing of English.

Reviewer 2 Report

The authors developed a new adaptive method based on carbon fiber reinforced plastics (CFRP) to reduce the thermal deformation of ball screws and improve its accuracy. Based on the experimental study, the adaptive method was verified. Specifically, the thermal deformation reduction can reach up to 70.1%, suggesting the effectiveness in reducing the thermal deformation of ball screws of the proposed method. The work is interesting and can be published in Materials if the following issues can be addressed:

1- The authors should cite the book chapter “Direct spinning of horizontally aligned carbon nanotube fibers and films from the floating catalyst method” conducted by Hai M. Duong et al. (Nanotube Superfiber Materials, 2019, William Andrew Publishing) and “Effect of alignment and packing density on the stress relaxation process of carbon nanotube fibers spun from floating catalyst chemical vapor deposition method” conducted by Hamed Khoshnevis et al. (Colloids and Surfaces A: Physicochemical and Engineering Aspects, 2018, 558, 570-578) in the introduction section for better reviewing the application of carbon based fiber composite for ball screws.

2- In section 3.1, why did the authors choose the rotational speed of 800 r/min for the numerical simulation?

3- In page 3, lines 111-114: why was the heat generation from supporting bearings ignorable and how did this affect the accuracy of the results?

4- Did the simplification of the geometric modelling features reduce the accuracy of the simulation results?

5- In page 6, lines 185-187: is it possible to include the heat generated in the motor, and other mechanical friction to the simulation work?

6- In Figure 14, why was the difference between FR result and Kriging result for sample 4 much higher than the others?
